# Understanding the implementation of continuity-enhancing innovations as steps towards midwife-led continuity of care: A qualitative study using Normalization Process Theory

Renate Simmelink[1,2,3,4]☉*, Anneke Pouwels[5]☉, Marianne Nieuwenhuijze[5,6], Arie Franx[7], Hanneke Harmsen van der Vliet – Torij[8], Ank de Jonge[1,2,4,9], Corine Verhoeven[1,2,4,9,10], Nadine van der Lee[11]

1 Amsterdam UMC location Vrije Universiteit Amsterdam, Midwifery Science, Amsterdam, The Netherlands, 2 Midwifery Academy Amsterdam Groningen, InHolland, Amsterdam, The Netherlands, 3 Amsterdam Public Health, Quality of Care, Amsterdam UMC, Amsterdam, The Netherlands, 4 Department of Primary and Long-term Care, University of Groningen, University Medical Center Groningen, Groningen, The Netherlands, 5 Research Centre for Midwifery Science, Zuyd University, Maastricht, The Netherlands, 6 Care and Public Health Research Institute (CAPHRI), Maastricht University, Maastricht, The Netherlands, 7 Department of Obstetrics and Gynecology, Erasmus MC, Rotterdam, The Netherlands, 8 Rotterdam University of Applied Sciences, Research Center Innovations in Care, Rotterdam, The Netherlands, 9 Amsterdam Reproduction and Development, Amsterdam UMC Location VUMC, Amsterdam, The Netherlands, 10 Department of Obstetrics and Gynecology, Maxima Medical Center, Veldhoven, The Netherlands, 11 Department of Obstetrics and Gynecology, Alrijne Hospital, Leiderdorp, The Netherlands

☉ These authors contributed equally to this work
* r.simmelink@amsterdamumc.nl

## Abstract

### Objective

To examine how specific midwife-led continuity of care elements are implemented and sustained in a maternity care system where independent community midwives collaborate with hospital-based care professionals.

### Methods

An explanatory qualitative study was conducted, using the Normalization Process Theory as a conceptual framework. Maternity care networks that had implemented an innovation contributing to midwife-led continuity of care were included in the study. Stakeholders invited to participate in semi-structured interviews included community midwives, hospital-based midwives, obstetricians, managers, an obstetric nurse, and healthcare insurance company employees. Participants were recruited through an initial purposive sampling strategy. As data collection progressed, theoretical sampling was applied.

### Results

A total of 47 interviews were conducted with stakeholders from nine different maternity care networks. While many participants expressed strong conceptual support for

**Data availability statement:** Data cannot be shared publicly because they consist of qualitative interview transcripts containing sensitive and potentially identifying information. Additionally, the data are in Dutch. Data are available from the Amsterdam UMC Medical Ethics Committee (contact via metc@amsterdamumc.nl) for researchers who meet the criteria for access to confidential data.

**Funding:** This study was based on data from the Continuity Of MIdwifery Care (COMIC) project, which was initiated by the Royal Dutch Organisation of Midwives (KNOV) and funded by ZonMw. The funders had no role in study design, data collection and analysis, decision to publish, or preparation of the manuscript.

**Competing interests:** The authors have declared that no competing interests exist.

midwife-led continuity of care (coherence), this did not always translate into aligned action in practice (collective action). Trust between stakeholders and financial feasibility were dominant themes across all four constructs of the Normalization Process Theory. Trust was found to develop incrementally through small collaborative steps but remained fragile in competitive or hierarchical networks. Financial structures shaped both engagement and resistance.

## Conclusion

The implementation process and normalization of midwife-led continuity of care elements are not only shaped by practical or organizational considerations but are also highly dependent on trust between stakeholders and financial feasibility. As dual preconditions, trust and financial alignment must be in place for stakeholders to consider new collaborative models and to be able to act in line with their beliefs. In systems with longstanding institutional divisions and fragmented funding, transformative change requires simultaneous investment in relational infrastructure and financial redesign.

## Introduction

Midwife-led continuity of care (MLCC) models, in which a known midwife or small group of midwives provides care throughout the antepartum, intrapartum, and postpartum periods, in collaboration with obstetric teams when required, are widely recognized for their benefits. The World Health Organization (WHO) recommends MLCC in settings with well-functioning midwifery programs, emphasizing the importance of relational continuity between women and their midwives [1]. While MLCC models are implemented in various ways globally, they share a common goal: fostering a trusting relationship between women and their midwives across the maternity care continuum [2].

Extensive research has demonstrated that MLCC improves maternal and neonatal outcomes. Compared to other models of care, MLCC is associated with higher rates of spontaneous vaginal birth, and lower rates of caesarean section, assisted vaginal birth, and episiotomy [2,3]. Additionally, women receiving MLCC report greater satisfaction due to the personalized care, empowerment, and trust, fostered through the relationship with their midwives [4]. Despite these benefits, the implementation and normalization of this care model is challenging [5].

The MLCC model is a complex intervention as it consists of multiple interacting components, affects various stakeholders and organizations, is context-dependent, and requires both behavioral, social and system-level changes for effective and sustainable implementation [6,7]. Given this complexity, an immediate and complete shift to MLCC is unfeasible, and implementation often occurs incrementally. Internationally, various healthcare innovations aim at improving continuity of care. However, challenges persist in embedding MLCC into routine practice and sustaining its integration within diverse social and organizational contexts. Key mechanisms and

contextual factors influencing the success of MLCC implementation in high-income countries were identified in a realist review [6]. In this review, the role theory and power theory were used as analytical frameworks to examine how contextual structures and professional dynamics shape implementation outcomes [8,9]. While these perspectives offer valuable insights, they are not specifically designed to explain how interventions become embedded in routine practice. To build on these findings and generate actionable, practice-oriented insights, it is important to apply an implementation theory that specifically addresses the processes by which new practices are adopted, enacted and sustained.

Normalization Process Theory (NPT) is an implementation theory that offers a complementary lens by focusing on the social processes and the work people do to implement and sustain MLCC initiatives in maternity care systems [10,11]. It states that successful implementation of an intervention into routine care depends on the collective efforts by individuals and organizations, ensuring that an intervention becomes normalized in daily practice. The NPT framework focuses on four key mechanisms: coherence (sense-making), cognitive participation (engagement), collective action (the work done to enable implementation), and reflexive monitoring (the formal and informal appraisal of benefits and costs) [10]. Together, these mechanisms help to uncover how stakeholders make sense of, engage with and operationalize MLCC-enhancing initiatives in practice, thereby normalizing it in daily routine care. How these mechanisms operate in practice is shaped by the specific context in which MLCC implementation takes place.

In the Netherlands, the organization of maternity care presents context-specific conditions that influence the normalization of MLCC. The Dutch maternity care system is divided into primary care, which is provided in the community by independent midwives, and secondary and tertiary care, which is provided in hospitals by hospital-based midwives, nurses and obstetricians [12]. Allocation to these tiers is based on the individual care needs of each woman. Women with a low-risk pregnancy receive care from independent community midwives and have the option of birthing at home, in a birth center, or to have a midwife-led hospital birth. Women who develop an intermediate or high risk for complications are referred to obstetrician-led hospital care, often resulting in a change of care professionals [13]. Such referral systems can disrupt continuity of care, as women often need to transition between different care settings and professionals throughout their pregnancy, during childbirth, or in the postpartum period. Therefore, the aim of this study was to examine how specific midwife-led continuity of care elements are implemented and sustained in settings where independent community midwives work in collaboration with hospital-based care professionals.

## Organization of maternity care in the Netherlands

Maternity care in the Netherlands is organized in a fragmented system with a division between primary care provided in the community and secondary and tertiary care in hospitals [12]. Women's care is based on the assessment of the individual risk of each woman. Women with an uncomplicated pregnancy receive care from independent community midwives in midwife-led, primary care, and have the option for home-birth. Most midwives work in group practices, of which the size can vary [14]. These midwives are the lead professionals during the whole childbirth continuum, as long as no medical risk factors arise. Women at intermediate or high-risk are referred to obstetrician-led secondary or tertiary care where they are looked after by a multidisciplinary team, including hospital-based midwives, obstetricians and obstetric nurses [13]. Obstetric nurses provide supportive clinical care and do not have the same autonomous responsibilities as midwives. Maternity care networks are a formal regional partnership between a hospital and the surrounding community-based midwifery practices. Within a maternity care network, hospital-based care providers and community midwives work closely together to coordinate maternity care, ensuring clear referral pathways, and shared responsibility throughout pregnancy, childbirth, and the postnatal period. Community-based midwifery practices may be affiliated with more than one maternity care network.

The Dutch maternity care system is financed by health insurance companies, regulated by the government. They have a considerable influence on determining which innovations or organizational changes are supported or reimbursed. Currently, the financial system for primary care and hospital care is separate and allows different reimbursement models. Community midwives receive a fixed fee per care episode per client (pregnancy, intrapartum care, postpartum care) and

hospitals are reimbursed based on 'diagnosis-treatment combinations (DTCs)'. These DTCs are based on the type of services, treatments and procedures that will be carried out, meaning they can be seen as fee-for-service grouped into care episodes [15,16].

## Methods

### Design

We conducted an explanatory qualitative study within an interpretive paradigm, using Normalization Process Theory (NPT) as a conceptual framework to examine the implementation and normalization of MLCC [10,11]. This study was part of the Continuity Of MIdwifery Care (COMIC) project, which aims to gradually implement MLCC within the Dutch maternity care system (S1 File). A realist review conducted as part of this project identified how and under what circumstances MLCC can be implemented [6]. To complement this theory-driven understanding, we used NPT to guide the current study adding a focus on the implementation process and social processes involved in embedding MLCC into routine care. This study is reported according to the Standards for Reporting Qualitative Research (SRQR) checklist (S2 File) [17].

### Setting, participants and data collection

Maternity care networks, including both Integrated Maternity Care Organizations (IGOs) and Maternity Care Networks (VSVs), that implemented an MLCC-enhancing innovation, were included in the study. These innovations provided more women the opportunity to continue receiving care from their own known midwifery team, instead of a complete transfer to obstetrician-led care. Maternity care networks are regional collaborations between hospital-based and community-based care professionals, aimed at organizing and coordinating perinatal care in a defined region. IGOs are formal partnerships with shared organizational and financial responsibilities, whereas VSVs are primarily collaborative networks without formally shared budgets or governance. These networks were identified and recruited through the COMIC research team's professional network, ensuring a diverse sample across the Netherlands. We included networks that implemented specific MLCC contributing innovations related to either gestational diabetes management or induction of labor with a Foley catheter at 41 weeks in community midwife-led settings. In these regions, this care was previously provided exclusively in hospital settings but is currently shifting to community midwife-led or shared care. Additional to these focused interventions, we included networks that made broader organizational changes to achieve a higher degree of continuity. One network enhanced continuity by shifting care from obstetrician-led to midwife-led models. Besides gestational diabetes management in midwife-led care and induction of labor with a Foley catheter at home, in this network all first consultations were relocated to primary care regardless of women's risk profiles. In other networks, continuity was facilitated by midwives working across both community and hospital settings. These cross-setting midwives were able to continue providing care after a referral from midwife-led to obstetrician-led care.

A total of 53 stakeholders from nine maternity care networks across the Netherlands were invited to participate. Stakeholders invited to participate were community midwives, hospital-based midwives, obstetricians, managers, and healthcare insurance company employees. The networks represented both highly urbanized and less urbanized regions and were affiliated with general hospitals or general teaching hospitals, ensuring a broad representation of maternity care contexts. In the Dutch healthcare system, hospitals are typically classified into three levels: general hospitals, general teaching hospitals and academic hospitals, based on the complexity of care provided, their educational role, and their involvement in research.

Participants were recruited through an initial purposive sampling strategy. The first participants invited were members of regional MLCC implementation project groups, as they were directly involved in the design and implementation of the innovation. As data collection progressed, theoretical sampling was applied, meaning that additional participants were selectively invited based on emerging findings from earlier interviews. By iteratively refining participant selection, key

perspectives and experiences relevant to the implementation of MLCC were deeper explored. Sampling continued until sufficient information had been obtained within each region to capture experiences relevant to the constructs of NPT, and no substantively new insights emerged in subsequent interviews. Participants were invited by email, telephone or in person. Semi-structured interviews were conducted and recorded via Microsoft Teams by RS and AP, using a topic list based on sensitizing concepts considered relevant in literature (S3 File). Transcripts were anonymized and stored in a secure environment. All participants provided written informed consent prior to participation. Participants were informed about their right to withdraw at any stage without consequences.

### Data analysis

The interviews were analyzed using a Thematic Framework approach [18]. Audio-recordings were transcribed verbatim, after which we familiarized ourselves with the data through repeated readings of the transcripts. In our study, the four core constructs of NPT served as the initial deductive analytical framework: coherence (sense-making), cognitive participation (engagement), collective action (work done to enable implementation), and reflexive monitoring (formal and informal appraisal of benefits and costs) [10]. Coding was conducted using this framework, while simultaneously allowing for inductive identification of themes not captured by NPT. The new analytical framework was systematically applied to all transcripts, assigning data segments to the relevant NPT constructs and newly identified themes. To ensure rigor in the analysis, RS, AP and NvdL, who have experience as a hospital-based midwife, a community midwife, and an obstetrician, respectively, independently coded and thematized three interviews. Subsequently, RS, AP and NvdL thoroughly discussed coding decisions and refinement of the analytical framework until consensus was reached. The remaining interviews were coded by RS and AP. Regular team discussions enhanced reflexivity and maintained coding consistency. Relationships between the emergent themes and the four NPT constructs were visualized in a conceptual figure. MAXQDA software was used for coding and data organization.

### Ethics

The Medical Ethics Committee (METC) of Amsterdam UMC declared that no formal ethical approval was required for conducting this study (Ref. No. 2022.0382).

## Results

Between July 2022 and May 2024, 47 interviews were conducted by RS and AP across 9 regions. Table 1 provides an overview of the key characteristics of the participating regions, including the implemented types of MLCC related innovations and the organizational structure (type of Maternity Care Network) in which these innovations took place. The interview duration ranged from 25 to 75 minutes. The experiences of community midwives (n = 17), obstetricians (n = 9), hospital-based midwives (n = 7), managers (n = 6), healthcare insurance company employees (n = 4), and an obstetric nurse (n = 1) were explored. Additionally, three midwives were practicing across both primary and secondary care settings, allowing them to continue providing care when a woman was referred to hospital-based care. Six of the eligible professionals who were invited did not participate due to being not able or willing to participate (n = 2) or not responding (n = 4). Baseline characteristics of the participants are shown in Table 2.

Trust and financial feasibility emerged as central and cross-cutting themes overarching the four core constructs of NPT, influencing the implementation and normalization of MLCC, see Figure 1 (S1 Fig).

### Coherence – Making sense of MLCC

Recognizing and believing in the benefits of MLCC promotes implementation. Many participants expressed strong conceptual support for MLCC driven by the intention of a client-centered approach.

**Table 1. Characteristics of the regions, n = 9.**

| Region | Type of Maternity Care Network (IGO/VSV) | Type of hospital | MLCC related innovation |
|---|---|---|---|
| A | Maternity Care Network (VSV) | general hospital | induction of labor with a Foley catheter at home |
| B | Maternity Care Network (VSV) | general teaching hospital | induction of labor with a Foley catheter at home |
| C | Maternity Care Network (VSV) | general teaching hospital | gestational diabetes management in midwife-led care + induction of labor with a Foley catheter at home |
| D | Maternity Care Network (VSV) | general hospital | gestational diabetes management in midwife-led care |
| E | Integrated Maternity Care Organization (IGO) | general teaching hospital | gestational diabetes management in midwife-led care |
| F | Maternity Care Network (VSV) | general teaching hospital | gestational diabetes management in midwife-led care |
| G | Maternity Care Network (VSV) | general hospital | cross-setting midwives* |
| H | Integrated Maternity Care Organization (IGO) | general teaching hospital | multiple care shifts from obstetrician-led to midwife-led care** |
| I | Maternity Care Network (VSV) | general hospital | cross-setting midwives* |

*midwives practicing across both primary and secondary care settings.

**gestational diabetes management in midwife-led care + induction of labor with a Foley catheter at home + all first consultations relocated to primary care regardless women's risk profiles.

**Table 2. Characteristics of the participants (n = 47) across the regions.**

| Characteristics | Region A | Region B | Region C | Region D | Region E | Region F | Region G | Region H | Region I | Total (n) |
|---|---|---|---|---|---|---|---|---|---|---|
| **Age (in years)** | | | | | | | | | | |
| **<30** | | | 1 | 1 | | | | | | 2 |
| **30-39** | 2 | 3 | 3 | 1 | 2 | 1 | 1 | 3 | 2 | 18 |
| **40-49** | | 1 | 2 | | 1 | 1 | 1 | 4 | 2 | 12 |
| **≥50** | 3 | 2 | 2 | 1 | 2 | 2 | 1 | 2 | | 15 |
| **Gender** | | | | | | | | | | |
| **Male** | | | 2 | | | 1 | | 3 | | 6 |
| **Female** | 5 | 6 | 6 | 3 | 5 | 3 | 3 | 6 | 4 | 41 |
| **Function** | | | | | | | | | | |
| **Community midwife** | 2 | 2 | 3 | 2 | 2 | 1 | 1 | 3 | 1 | 17 |
| **Hospital-based midwife** | 1 | 2 | 1 | | 1 | 1 | | 1 | | 7 |
| **Cross-setting midwife*** | | | | | | | 2 | | 1 | 3 |
| **Obstetrician** | 1 | 1 | 1 | 1 | 1 | 1 | | 2 | 1 | 9 |
| **Manager** | 1 | | 2 | | 1 | 1 | | 1 | | 6 |
| **Obstetric nurse** | | | | | | | | 1 | | 1 |
| **Healthcare insurance company** | | 1 | 1 | | | | | 1 | 1 | 4 |
| **Number of years in practice** | | | | | | | | | | |
| **0-9** | 1 | 2 | 2 | 2 | 1 | | | 2 | 1 | 11 |
| **10-19** | 2 | 2 | 2 | | 2 | 2 | 2 | 4 | 3 | 19 |
| **≥20** | 2 | 2 | 4 | 1 | 2 | 2 | 1 | 3 | | 17 |

*midwife practicing across both primary and secondary care settings.

*"How I would see it ideally is that you, in a safe and pleasant way, regardless of the money, can take an ideal look at what kind of care is best for the client. Please, put the client at the center. For a moment, don't look at the finances, or at what suits you better or what suits someone else better, but please, first look at the client." (Community midwife, Region A)*

In various regions, the implementation of MLCC innovations is often guided by a project group consisting of representatives of different healthcare disciplines. Because implementation can result in significant changes to current care or current care processes, participants emphasized that it is important not only for the project group members, but for all involved stakeholders at all organizational levels, to recognize the benefits of MLCC. Participants report that pursuing MLCC was not always the primary objective. Healthcare professionals in the community and the hospitals wanted to support each other in particular when it comes to innovations that address capacity issues and to cope with the increase in inductions of labor. The success of the implementation of innovations also appears to depend on the initiator, and more specifically the hierarchical position of the initiator in some regions. Additionally, it seemed helpful if community midwives were united in this regard and presented a shared, cohesive voice for a more effective external representation in discussions with other professionals.

### Cognitive participation – Engagement in MLCC

Participants stated that a shared vision promotes the implementation of MLCC. However, in reality, it is challenging to align visions between community and hospital-based care professionals but also between and within community midwives, and between and within hospital-based care professionals in the same region. Some participants reported that other hospitals in the same region were not willing to participate because the innovation might contribute to more unplanned care and because of a different vision on the innovation. A lack of support among hospital-based care professionals for the shift in care towards midwife-led settings, makes transitions unlikely to occur. The engagement of the obstetrician appears to be conditional.

> *"A challenge that actually arises everywhere when it comes to shifting care, […] but there needs to be support regardless, including from the obstetrician. And if that support from the obstetrician is lacking, then you often see that it ultimately doesn't lead to a certain shift that is indeed necessary and perhaps entirely feasible. And that's an area where I think secondary care should also play a role […]. And that's partly about letting go, and maybe it's also partly about fear, a culture of fear, about having trust and certainty like "okay I have my own patient population that I currently see, but I'm able to let go of that population and trust that they can also be properly and adequately cared for in community care"." (Healthcare insurance company employee)*

When visions are aligned, various aspects (logistics, finances, organizational structure) nevertheless can cause this vision to falter in practice. Some community midwives reported that they only felt the motivation to engage when the innovation was a complete shift in care rather than only a part of it. For example, requiring women to go to the hospital for the insertion of the Foley catheter disrupts MLCC and calls into question whether this approach truly results in more continuity and cost-effective care. However, according to most participants, implementing parts of MLCC gradually can contribute to trust between healthcare professionals and thereby ultimately expanding the innovation.

### Collective action – Work done to enable implementation of MLCC

For engagement within the innovation, it is important that all professional groups are represented within the project group in a well-balanced composition. According to participants, this approach fosters shared ownership of the innovation, and is essential for both successful adaptation and efficient implementation. For instance, some participants noted that obstetric nurses are sometimes overlooked despite their crucial role in the execution phase.

Participants expressed the need for a well-defined project plan to guide the implementation of the innovation. It is also beneficial for implementation if the project group has an independent, external leader especially when there are differences in priorities, concerns, objectives and interests among the stakeholders. Several participants described concerns about ambiguity in responsibilities, defining professional boundaries and task divisions. Participants do not always

perceive support from professional associations. In order to make progress at the micro level, they express the need for established structural support at the macro level.

*"From the national organizations, even just jointly developing protocols and guidelines. If they come up with innovations, then you would already expect them to collaborate on that. And as long as they don't... If you're talking about antenatal CTG [cardiotocography] in community care, just as an example […]. It's all well and good that the KNOV [professional association of midwives] comes up with that, and that they even ended up getting approval from the NZA [Dutch Healthcare Authority], because they say, "Yes, this is indeed a good idea". But if KNOV had already coordinated this from the start with the NVOG [professional association of obstetricians]... but maybe that wasn't feasible either, right? Because even there you have completely different visions. So it's basically the same thing that happens on a small scale within a VSV; the same thing happens nationally with those professional associations. If you had already started that jointly, drafted it together, and thought about: how can we actually make this possible together." (Manager, Region C)*

On the one hand there is a recognized need to formalize responsibilities in a clear and unambiguous manner. On the other hand, healthcare also requires flexibility to achieve a woman-centered approach.

### Reflexive monitoring – Evaluation of MLCC implementation

Evaluation is often based on short-term financial and organizational impact and clinical outcomes rather than long-term health benefits or satisfaction of both clients and healthcare professionals. According to the participants, the most prominent reason for the discontinuation of the innovation was the lack of clearly defined evaluation criteria beforehand, particularly the lack of a well-defined definition of what would be considered a successful implementation. For example, cross-setting midwives are assumed to promote continuity by bridging settings. However, participants noted that in practice this often came at the expense of one setting, particularly community care, resulting in more medicalized care and a higher rate of interventions.

*"Too quiet to have two midwives on duty, but sometimes too busy to manage it alone, [...] and then you simply don't have time anymore to sit with your woman in labor. […] Because, you just walk in, and purely look at: how far has this woman progressed? Just what you can measure, via vaginal exams, but not really looking at her or maybe not even really listening to her. And then things escalate quickly: okay, let's give her a sniff of oxytocin, and before you know it, it becomes medical. Because the threshold is low, everything is within reach. So, there's no delay in that sense. Very quickly you'd connect a CTG. And that tendency, sometimes maybe to make things medical even when it isn't truly necessary." (Cross-setting midwife, Region I)*

In some regions, the success of MLCC implementation was measured primarily by an increase in births within community care settings. Participants noted concerns that this outcome measure does not fully capture the impact of the MLCC innovations. Recommendations were made to assign greater emphasis on for example hospital length of stay and satisfaction of women and healthcare professionals.

### Trust as a requirement for MLCC implementation

Trust between stakeholders, particularly between community and hospital-based care professionals, was often a prerequisite for even considering a shift in care responsibilities, and continued to play a crucial role in the whole normalization process. In regions where trusting relationships were described, stakeholders more readily engaged in collaborative efforts, enabling community midwives to take on a broader scope of practice. By contrast, in regions where trust was fragile or lacking, both structural and relational barriers hindered progress toward MLCC.

Importantly, trust was not always assumed/present, but had to be earned or demonstrated over time. Several participants described how care shifts were only made possible through incremental steps, where each phase had to meet the expectations of the involved parties before further expansion could be negotiated. In some cases, the willingness to transfer responsibilities to community midwives was conditional. This conditional trust underscores how perceived competence played a vital role, and stakeholders needed to feel confident not only in each other's clinical skills but also in their professional judgment and intentions in order to relinquish certain aspects of healthcare they were accustomed to delivering.

*"That collaboration really has existed for a very long time. I think what also characterizes it is that we always stay in conversation with each other. So we inform each other, back and forth, we are open with each other. Which also means that you just know much more about each other. What do you do? What are you working on? Which gives insight into each other's expertise. [...] What we're going to organize is a moral deliberation. Just to really talk with each other about: how do we deal with this, what do we think about it, what are our standpoints? Just so you know where the other person stands, what their perspective is. We're doing this together with both primary and secondary care. So we're really engaging in good dialogue together, and I think that's very important in our collaboration." (Manager, Region H)*

Trust was seen as a foundation for equivalent partnership. It was characterized as an ongoing process that requires sustained attention, especially because there is always some degree of turnover among healthcare professionals in the region. In regions where trust was described, healthcare professionals emphasized the importance of attitude, trust in physiological childbirth, and the intrinsic motivation to collaborate. Having experience of working in both community and hospital settings was expressed as helpful in fostering mutual understanding. In regions where trust was described, they expressed the belief that replacement of certain individuals by persons with a different vision on MLCC would not lead to substantial changes in established practices. Participants underscored the significance of being acknowledged and taken seriously and respected in their capacity as a professional group. In regions where unequal partnerships were described, professionals were more likely to adopt either a more dominant or a more subordinate position. Participants expressed concerns that in those regions the ability to discuss innovations that concerned a shift in care was often dependent on the influence of the most dominant actors.

*"Well, I really appreciate that it's possible and, uh, allowed. It also gives clients a lot of peace of mind that they're allowed to start labor with us, and that they have a better chance of getting those contractions going [...]. But a lot of clients then, well, according to the hospital, didn't meet the criteria. And sometimes we do think that they do meet the criteria. But, yeah... we don't go against that, of course. If they think, that the [fetal] head isn't engaged yet, while we felt in the practice that it was engaged and you couldn't push it back anymore, then that's quite unfortunate. But you don't go into discussions about it, and certainly not over the client's head." (Community midwife, Region A)*

Moreover, several stakeholders described how trust was influenced by unspoken doubts, such as the suspicion that financial incentives might drive decision-making. Furthermore, different visions on MLCC among community midwifery practices and a lack of cohesion between midwifery practices sometimes led to fragmentation and competition, further complicating efforts to implement MLCC.

## Financial impact of MLCC

The implementation of MLCC, along with the associated shift in care, carries significant financial implications. Ultimately, it may lead to either increased or decreased income alongside varying workloads across different levels of care. In order to realize financial compensation, a restructuring of the existing financial system seems required according to various stakeholders. Participants expressed the need for greater flexibility in billing for care services through smaller, more granular units, instead of the larger bundled payments that are frequently applied currently in the Netherlands. According to a

healthcare insurance company employee gradual care shifts appear to be not only crucial for fostering trust, but also from a financing perspective. Since shifts in income may impact the financial sustainability of various stakeholders within the healthcare system, they can potentially threaten their continued existence. However, shifts in care can be financially supported when appropriate bundled payment structures are available. According to participants, it is not sustainable that care that has been transferred to the community care setting also remains funded within the hospital care system. Stakeholders expressed that financial compensation is often lacking in terms of extra workload when shifting care to community settings. Hospital-based care professionals frequently experienced a decline in income. This seems to affect the level of engagement more strongly in certain regions than in others. In regions where this issue seems less prominent, it appears that hospital-based professionals are not that concerned about a potential decrease in workload, resulting in reduced income, due to the substantial demand for obstetric and gynecological services. However, participants in these regions likewise emphasized the need for future financial compensation for the additional time demands imposed by care shifts in combination with the associated loss of income. For that reason, financial considerations increasingly hinder high-value care.

*"There's definitely fear that it will impact our income, and that's not just for us, because primary care is afraid of that too, right? That they'll lose things and that it'll be felt in their wallet [...]. We've already made care shifts, like all first consultations in primary care, without considering the financial consequences. Yes, we never negotiated that with the healthcare insurer, that was really stupid [...]. We've always prioritized quality, but from now on we have to make sure that if care shifts, something must come in return from the healthcare insurance company. That we're not just going to invest time without getting anything in return, even though the quality of care is definitely improving." (Obstetrician, Region H)*

Participants repeatedly emphasized that when it comes to finances, trust is extremely important. Being able to build on pre-existing trust is expressed as beneficial. Participants noted that the historical context of collaboration may continue to influence regional dynamics over time. Nevertheless, even in the regions with pre-existing trust, financial matters occasionally led to friction. In this context, healthcare professionals highlighted the significance of exercising patience and maintaining an open and ongoing dialogue among alle parties involved. Participants noted that failing to do so opens the door to assumptions among healthcare professionals about possible financial interests or financially driven decision-making, which may strain rather than strengthen collaboration and therefore influence engagement. These suspicions were rarely addressed in open dialogue, according to participants.

*"But it has now been decided, as kind of compromise, especially for secondary care, that they still really want a video consultation for everyone with an abnormal glucose tolerance test. Fill in the blank: they get to open a DBC [Diagnose-Treatment-Combination] without actually needing to use it. So we do the work without DBC, and the obstetrician still gets their DBC registered. Of course, they frame it differently, but it's completely illogical to do a video consultation in this situation. So that will be our next evaluation point: that it's complete nonsense." (Community midwife, Region F).*

The transition to integrated funding is identified as challenging. According to a healthcare insurance company employee this requires effective collaboration, robust governance structures and adequate engagement of the full spectrum of healthcare professionals. Stakeholders, including a healthcare insurance company employee, described an optimal situation in which the volume-based incentives are eliminated, thereby enabling a true emphasis on providing the most appropriate and client-centered care.

## Discussion

In this study, the NPT framework was used to examine how MLCC elements were implemented and sustained across different maternity care networks with independently practicing community midwives in collaboration with hospital-based professionals. Our findings demonstrate how trust between stakeholders and financial feasibility were described as

cross-cutting and foundational mechanisms that can enable or hinder progress towards MLCC, and influence all four NPT constructs on micro, meso, and macro level. Trust between care professionals was described by participants as neither binary or static, but as something negotiated over time through small changes. Such incremental approaches were reported to contribute to growing confidence in each other's competencies, to support broader engagement, and to gradually extend the scope of MLCC. However, financial arrangements were described as a powerful influence on the ability to normalize MLCC, either facilitating or obstructing implementation efforts. Participants emphazised that transparent and equitable funding is essential to ensure that financial incentives do not undermine shared goals.

Initiatives that foster joint ownership, such as inclusive project groups and shared decision-making structures, were experienced as strengthening commitment and alignment among all stakeholders. Yet, the absence of clear, jointly a priori defined success criteria was perceived to hinder reflective learning and, in some networks, led to premature discontinuation. To support long-term sustainability, participants suggested that structured evaluation, encompassing clinical and financial outcomes, women's experiences, and professional perspectives, would be necessary. Furthermore, relying solely on individual maternity care networks to develop and sustain MLCC innovations was perceived as inefficient and inequitable. Participants called for stronger national coordination and support, including clear guidance on governance and financial models.

### Implications for practice

This study shows that trust between stakeholders and financial feasibility are not merely contextual factors, but conditional mechanisms that operate across all four NPT constructs, shaping the entire process of MLCC implementation. Although NPT has been widely recommended and applied to understand the implementation of complex interventions [19,20] and has been used to examine the implementation of continuity of carer in the UK [11], these applications did not explicitly conceptualize trust between stakeholders and financial feasibility as conditions that influence all constructs. Therefore, our findings extend existing NPT research by demonstrating whether and how relational and financial infrastructures shape coherence, cognitive participation, collective action, and reflexive monitoring can be activated and sustained in the implementation of MLCC enhancing innovations.

Where implementation studies have acknowledged the relevance of trust and financial feasibilities [21–23], our findings suggest that they are foundational and define the possibilities to implement MLCC enhancing innovations. For example, Metz et al. identified the development of trusting relationships among stakeholders as an important strategy related to the enrichment of implementation practice [21]. However, our study highlights trust as a conditional mechanism. This aligns with insights from interprofessional collaboration research, which suggests that relational trust is a precondition for equal decision-making and joined engagement [24]. However, trust does not arise spontaneously and must be actively developed. Many implementation efforts tend to focus on defining professional boundaries, task divisions, and responsibilities [25,26]. Yet, this study shows that sustainable change requires more than structural clarity, but also investment in relationships, mutual understanding and a shared willingness to renegotiate professional roles. In line with our findings, other research has shown that trust-building interventions like interprofessional training and interprofessional diversity during the course of study, regular face-to-face meetings, informal exchanges outside of work, and making an independent person responsible for overseeing the collaborative process may therefore be as critical as the design of a new care pathway [24,27,28].

In parallel, financial structures shape stakeholder engagement. In systems based on fee-for-service payment models, like in the Netherlands, MLCC often implies a redistribution of tasks without a corresponding redistribution of funds [6]. The financial burden of MLCC implementation often falls on the shoulders of the care professionals or institutions driving the change, while the financial benefits occur elsewhere in the system, like insurance companies. For example, a budget impact analysis on antenatal CTG in midwife-led care, leading to more continuity of care, showed that implementation would lead to increased operational costs for midwifery practices, but would lead to substantial cost savings for insurance

companies and society [29]. While previous studies identified reimbursement models as barriers to integrated or continuity-based care, they are often seen as external contextual constraints [6,11,29]. Our findings extend this understanding by showing how financial systems interact with perceived risk and trust, thereby directly influencing one's willingness to engage in collective action. As such, financial fears can become entangled with perceptions of threat, leading to disengagement, erosion of trust, or resistance. Notably, participants rarely discussed these dynamics openly, but instead veiled in normative arguments about safety, quality or logistics. Without financial models that reward continuity, collaboration, and better outcomes rather than volume or interventions, MLCC normalization will remain vulnerable.

While stakeholders may intellectually support the principles of MLCC and have a shared understanding of its benefits (coherence), this does not always translate into aligned action in practice (collective action). Underlying these contemporary challenges appears to be a deeper layer of historical and structural inequality. The longstanding hierarchical divide between midwives and obstetricians, rooted in centuries of struggle over autonomy, authority, and legitimacy, continues to shape current interprofessional dynamics in maternity care [30]. In many maternity care networks, this history manifests in unequal power dynamics, where hospital-based care professionals often hold dominant positions in decision-making processes. In turn, our study shows that community midwives may adopt more dependent roles, thereby creating conditions facilitating current hierarchical structures. This dependency can be a result of structural constraints, but also due to a cautious or hesitant stance toward claiming professional space. These hierarchical behaviors can limit the collaboration between midwives and obstetricians, while research shows that non-hierarchical interprofessional groups are more productive, feel more committed, work more efficiently and productively [31].

The use of NPT in this study made it possible to unpack these misalignments at the micro level, exposing how deeply embedded structures and relationships influence not just what people do, but why they do, or not do, act in line with what they believe. Rather than viewing resistance towards MLCC as a lack of motivation or misunderstanding, NPT reveals how implementation is shaped by a complex interplay of sense-making, engagement, collective action and evaluation. Moreover, NPT enabled us to identify misalignments between coherence and collective action. This disconnect, where stakeholders may express support for MLCC but fail to act accordingly, proved especially salient in contexts with a lack of trust or financial risk. By conceptualizing trust and financial feasibility as conditional mechanisms operating across all four NPT constructs, this study contributes a more enrich full understanding of why MLCC efforts stall despite apparent consensus about their value. Therefore, successful implementation of MLCC requires more than the introduction of a new protocol or redistribution of tasks. Implementation efforts should include trust-building strategies like interprofessional training and shared reflection [24,32]. Assumptions should be made explicit through open discussions about financial arrangements and each professional's interests.

## Strengths and limitations

This study included a diverse range of stakeholders from multiple maternity care networks and offers in-depth insights into the implementation processes of MLCC elements. These findings are relevant for all international settings where midwives work as independent professionals in collaboration with hospital-based care professionals. The use of NPT enabled a structured understanding of implementation dynamics, especially with regard to the implementation process and the social processes involved in embedding MLCC enhancing innovations into routine care, while also revealing the important underlying and overarching mechanisms of trust and financial alignment. It has provided concrete insights into where implementation efforts get stuck and what is needed to make change sustainable.

The iterative sampling strategy allowed us to refine participant selection throughout data collection to capture diverse and relevant perspectives. While recognizing the pivotal role of financial structures in the implementation of MLCC-enhancing innovations, healthcare insurance company employees were recruited for participation. This offered valuable insights into the economic dimensions of maternity care. Credibility was further strengthened through the inclusion of a wide range of stakeholders across different professional groups and maternity care networks, enabling triangulation

of perspectives. In addition, the multidisciplinary composition of the research team facilitated critical reflection, thereby enhancing confirmability of the findings.

In addition to these strengths, this study has some limitations. Although we aimed for a broad representation of stakeholders, two obstetricians were invited but declined participation. This could potentially limit the credibility of results. However, given the total number of interviews conducted, the absence of two stakeholders is unlikely to have significantly influenced the overall findings. Based on the other interviews conducted in the same regions, it became clear that these individuals may not have been in favor of the implementation of MLCC-enhancing innovations. The specific reasons for their opposition remain unknown, as their voices are not represented in the data. Furthermore, the perspectives of women and client advocacy groups were not included in this study. However, the focus was on implementation processes rather than client experiences. In addition, we only included one obstetric nurse as a result of the iterative sampling.

By focusing exclusively on regions where some degree of MLCC implementation had taken place, important insights may have been missed from regions where such efforts were unsuccessful or never initiated, potentially limiting transferability. These contexts could possibly reveal additional barriers and blind spots that remain invisible in relatively successful settings. Future research could build on our findings by conducting a realist evaluation, exploring how the identified mechanisms operate in different circumstances, and further explore what helps or hinders implementation processes towards MLCC. This approach may also help foster broader engagement, as stakeholders become active contributors to the research process rather than solely participants. In addition, it would be helpful to examine the implementation of a full MLCC model, although this appears to be difficult to realize in practice.

## Conclusion

This study highlights that the implementation process and normalization of MLCC-enhancing initiatives are shaped not only by practical or organizational considerations, but is highly dependent on trust between stakeholders and financial feasibility. Simply believing in MLCC is not enough for its successful normalization. As dual preconditions, trust and financial alignment must be in place for stakeholders to consider new collaborative models and to be able to act in line with their beliefs. This challenges more linear models of implementation and suggests that in systems with longstanding institutional divides and fragmented funding, transformative change requires simultaneous investment in relational infrastructure and financial redesign.

## Supporting information

**S1 File. Supplement material 1 – COMIC project.**
(DOCX)

**S2 File. SRQR_Checklist.**
(DOCX)

**S3 File. Supplement material 2 – Interview guide.**
(DOCX)

**S1 Fig. Figure 1 – Financial system as a requirement and trust as a central theme.**
(PDF)

## Acknowledgments

We would like to thank all participating maternity care networks and the various stakeholders for their valuable time and contributions to this study. We sincerely thank Sonia Dalkin for her valuable consultation and thoughtful input.

## Author contributions

**Conceptualization:** Renate Simmelink, Anneke Pouwels, Marianne Nieuwenhuijze, Ank de Jonge, Nadine van der Lee.

**Data curation:** Renate Simmelink, Anneke Pouwels.

**Formal analysis:** Renate Simmelink, Anneke Pouwels, Nadine van der Lee.

**Investigation:** Renate Simmelink, Anneke Pouwels.

**Methodology:** Renate Simmelink, Anneke Pouwels, Marianne Nieuwenhuijze, Ank de Jonge, Corine Verhoeven, Nadine van der Lee.

**Supervision:** Ank de Jonge, Nadine van der Lee.

**Validation:** Renate Simmelink, Anneke Pouwels.

**Visualization:** Renate Simmelink.

**Writing – original draft:** Renate Simmelink, Anneke Pouwels.

**Writing – review & editing:** Marianne Nieuwenhuijze, Arie Franx, Hanneke Harmsen van der Vliet – Torij, Ank de Jonge, Corine Verhoeven, Nadine van der Lee.

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
