## [Decision Letter · Decision Letter 0]

17 Nov 2025

Understanding the implementation of midwife-led continuity of care using the Normalization Process Theory: a qualitative study

PLOS ONE

Dear Dr. Simmelink,

Thank you for submitting your manuscript to PLOS ONE. After careful consideration, we feel that it has merit but does not fully meet PLOS ONE’s publication criteria as it currently stands. Therefore, we invite you to submit a revised version of the manuscript that addresses the points raised during the review process.

We look forward to receiving your revised manuscript.

Kind regards,

Karolina Linden, Ph.D

Academic Editor

PLOS ONE

**Journal Requirements:**

3. In the online submission form, you indicated that your data is available only on request from a third party. Please note that your Data Availability Statement is currently missing a link to where data requests can be made. Please update your statement with the missing information.

**Additional Editor Comments:**

Thank you for submitting your manuscript, which has potential for publication but needs revisions before it can be accepted. Please address all reviewer comments, especially methodological concerns.

Reviewers' comments:

Reviewer's Responses to Questions

**Comments to the Author**

1. Is the manuscript technically sound, and do the data support the conclusions?

Reviewer #1: Yes

Reviewer #2: Yes

2. Has the statistical analysis been performed appropriately and rigorously?

Reviewer #1: N/A

Reviewer #2: N/A

3. Have the authors made all data underlying the findings in their manuscript fully available?

Reviewer #1: No

Reviewer #2: No

4. Is the manuscript presented in an intelligible fashion and written in standard English?

Reviewer #1: Yes

Reviewer #2: Yes

Reviewer #1: 3. The authors have described why data cannot be shared publicly due to sensitive and potentially identifying information, which makes sense.

Please see the same review comments in the attached document.

Peer-review of ”Understanding the implementation of midwife-led continuity of care using the Normalization Process Theory”

• Most important information

o Summary of the research and your overall impression

Thank you for the opportunity to review this paper using the Normalization Process Theory to understand the implementation of midwife-led continuity of care. The paper is well written, and using the NPT theory as a conceptual framework to understand how MLCC can become embedded and sustained in everyday work is a good approach. The results are significant for further implementation of MLCC. However, the authors need to clearly define that what you have assessed is not a full implementation of MLCC. While this nuance is mentioned in places- such as in the Supplementary material 1, where you note that ”some degree of MLCC had been implemented” – the introduction and methods sections currently give the impression that a full MLCC model was implemented.

o Major issues that must be addressed

The Introduction section gives an overview of the midwifery continuity of care (MLCC) model, but when reading the results I understand that this is more about task shifting or shifting care to community settings, and not really about implementing a MLCC model with a small group of midwives providing continuity 24/7 throughout the continuum of care with on-call for their women. The results are important and well-written but the Introduction needs to align better with the rest of the paper i.e. to focus more on midwifery led care and less on the results of the MLCC model (ref 2-4), as this is not the intervention you have explored. A MLCC model also have a strong midwifery philosophy with a focus on women-centred care and less use of unnecessary interventions, and I think the results do not align with the full concept of the model. I feel that using the word MLCC is not really correct when you only talk about part of this concept, i.e. shift in care towards midwife-led settings, and I wonder what the participants discuss when you describe that they had a strong conceptual support for MLCC.

In line 96-98, the authors describe how care is provided for low-risk women but it needs to be added the degree to which continuity is provided. Does the woman meet the same or a few midwives in this model?

In line 133-134, the authors need to clearly define what you mean with ”full continuity of care”.

It is not clear how healthcare is financed in the Netherlands, which makes it a bit confusing to read about financial aspects in the results. There is a short description of the financial system in the discussion (In systems based on fee-for-service payment models, like in the Netherlands,…) but this need to be described in the Introduction for the reader to understand the problems described in the Results.

It is also not clear what role health insurance companies have. Please describe this in the background/context.

In line 95, the authors also mentioned what professions are working in the Dutch maternity care system, but it would be helpful if you could describe what role the obstetric nurse has in relation to the midwife.

o Minor issues to address

Research question: Since the authors have only used one research question, I recommend changing it to an aim instead.

Methods

Design: Add a reference to the SRQR checklist.

Setting, participants and data collection: Line 123, …that implemented an innovation contributing to MLCC. The original sentence implies that the innovation directly advanced MLCC, but the results do not describe the extent of continuity. Did women meet a known midwife? I suggest changing this to focus more on the process of increasing care provided by midwives (in primary care), or to clearly illustrate how this innovation improved continuity of care.

Line 142, remove the comma after “participate”.

Please provide more details of the data collection process, how the participants were recruited, through email or phone?, whether the interviews were recorded (you only write about transcripts), and who conducted them.

Line 154-155, please give some suggestions on the questions or add a copy of the interview guide for the reader to better understand the concepts you were discussing with the participants.

Results

Please relocate Table 2 and the short description of the table to the Methods section, e.g line 148, and that Figure 1 be moved to the current position of Table 2, before describing the results. In addition, it would be beneficial to have a short description of the results before the constructs and teams are described, e.g moving the first part of the paragraph describing the theme in line 300-301 and adding something about financial aspects. ”Trust emerged as a central and cross-cutting theme overarching the four core constructs of NPT, influencing the implementation and normalization of MLCC (see Figure 1).”

This reordering would enhance the logic structure of the manuscript and improve coherence of the narrative flow.

I suggest writing the NPT construct before each theme, e.g Coherence- Making sense of MLCC, to facilitate the reader’s understanding.

Discussion

Trust and financial feasibility are discussed as fundamental for improving MLCC. This is well described and discussed with valuable insights. However, the discussion would be improved if you incorporated other references to it, as you have done in the Implications for practice section. Please revise these two sections (Discussion and Implications for practice) and merge them into one, including references of others’ work.

Strengths and limitations

The authors do not discuss the limitations of not examining the full implementation of a MLCC model, as only task shifting and shifting care to community settings are explored. Please add this as a limitation.

Line 448, please add “of” before the word trust (mechanism of trust).

Conclusion

The conclusion is well-written and provides a clear and concise summary of the findings, as well as what is required to strengthen the implementation of MLCC. However, I am hesitant about the phrasing of the first sentence (line 531-532), as the study does not examine the implementation of a full MLCC model. I recommend revising this paragraph to align with my overall comment on the manuscript, ensuring that the scope of the intervention is accurately represented.

Reviewer #2: Thank you for the opportunity to review this interesting and important study addressing the implementation of MLCC within existing care systems. This work contributes to the important body of knowledge on how evidence-based care can be provided to women. The manuscript is well written and employs Normalization Process Theory to explore the implementation process. The inclusion of multiple stakeholders offers a broad range of perspectives, which enriches the understanding of MLCC implementation.

A more detailed description of how the Framework Method was used in the analysis would have been beneficial. As a reader, I find it difficult to follow what the authors have done and how they have done it. Consequently, I consulted the cited reference for the method (14) to gain a basic understanding, as I am not familiar with it. In that paper, the method is described in six steps, and the analysis is conducted using a matrix. Did you use a matrix? Is it included anywhere in the submission? Would it make sense to publish it, or provide examples of how you worked with this matrix? Alternatively, is the NPT your matrix? Including such material might help readers follow the description and understand the details of the analysis.

I find it somewhat difficult to follow the description of the setting and to determine whether all stakeholders are involved in, or working closely with, a setting that provides MLCC. In the Methods section, under Setting, participants and data collection (line 123), the phrase “innovation contributing to MLCC”, and later (line 134) “networks that appeared to come close to achieving full continuity of care” and “enhanced continuity by shifting care from obstetrician-led to midwife-led models” seem rather vague. A model can be midwife-led and continuity can be increased without meeting the criteria for classification as an MLCC model (as outlined in your Introduction, line 55). Do you mean a modified MLCC model? Do these innovations indicate that MLCC was achieved, or were they part of an implementation process aimed at establishing an MLCC model?

In the Results section, under Financial impact of MLCC (line 358), the statement “The implementation of MLCC, along with the associated potential shift in care, carries significant financial implications” raises some similar questions. If MLCC is implemented, does this not necessarily entail a shift in care? Why is this described as a ‘potential’ shift?

Figure 1. is presented in the middle of the Results section. To provide an overview of your findings more effectively, it might be useful to present the figure earlier or to describe your findings in terms of themes, their names and interrelations. The current structure of the Results section introduces the reader directly to one of the four core concepts of the NPT. However, in Figure 1, these core concepts do not appear to interrelate as described in the cited framework. It might have been helpful to include an introduction explaining how the authors conceptualised the changed relationships of the core concepts and the introduction of the additional findings of financial system and organizational system surrounding the NPT core concepts and the central concept of ‘trust’. This is for example described with the following phrase in the Discussion, (line 149) ‘Our findings demonstrate how trust and financial feasibility can be seen as cross-cutting and foundational mechanisms …’

Additionally, could you elaborate on what you mean by the concept of “trust” referred to in this context? Is it trust between stakeholders and their collaboration, trust in the concept of MLCC, trust in the shift of care systems, or something else?

.

Reviewer #1: No

Reviewer #2: No

---

## [Author Response · Author response to Decision Letter 1]

15 Dec 2025

Reviewer 1

Thank you for your time and effort in reviewing our manuscript and providing valuable comments and suggestions, which helped us to improve the manuscript.

We agree that the original version of the manuscript could give the impression that full MLCC models had been implemented in the included networks. As suggested by the reviewer, we have now clarified throughout the manuscript that the study concerns innovations aimed at increasing continuity of care, rather than full implementation of MLCC. In addition, we changed the title and have added a sentence to the introduction to state that an immediate implementation of a complex intervention like MLCC is not feasible.

Please see page 2 lines 31 and 49, page 3 lines 72-74, page 4 line 92, page 6 line 148, page 7 line 157, page 22 lines 488 and 497, and page 25 line 568.

We agree that the innovations implemented in the included networks do not constitute a full MLCC model. We have carefully considered your suggestion to avoid the term MLCC, however, we respectfully believe that retaining MLCC as the overarching concept remains appropriate. We have seen that MLCC is a complex intervention that can only be implemented through incremental system changes. A new model needs to evolve through stepwise innovations, including task shifting and expanded midwife-led care. Therefore, the innovations studied here represent steps along the pathway towards MLCC. As stated above, we have clarified throughout the manuscript that our study focuses on continuity-enhancing innovations moving towards MLCC, rather than the implementation of a complete MLCC model. We hope the revisions make the alignment between the Introduction and Results clearer while allowing us to retain MLCC as the most appropriate overarching term and aim.

We agree that the original description of care for low-risk women did not sufficiently clarify the degree of continuity provided within the Dutch maternity care model. To address this, we have added an explanation in a textbox describing the Dutch maternity care system. In this textbox, we clarify that most community midwives work in group practices of varying size, with a supporting reference, see page 5 line 119. We believe this addition provides more clarity.

We agree that the phrase “full continuity of care” needs a clearer definition and may elicit questions. In the revised manuscript, we have adjusted this sentence to avoid implying the existence of a fully implemented MLCC model. Instead, we now specify that some regions had implemented broader organizational changes that increased continuity to a higher degree than the other specific innovations, see page 7 lines 160-162. The present description clarifies that these networks did not achieve full MLCC, but rather took more extensive steps toward improving continuity.

We agree that the original manuscript did not provide sufficient background on the organization and financing of Dutch maternity care to fully contextualize the findings noted in the Results. In response, we have added a textbox in the Introduction, see pages 5 and 6 from line 114, that explains:

- how maternity care is financed in the Netherlands for community midwives and the hospitals (lines 125-132);

- the role of health insurance companies (lines 126-127);

- the role of the obstetric nurse in relation to the midwives (lines 123-124).

We have revised the research question into an explicit aim in the Introduction, see page 5 lines 105-110.

Methods

We have added the supporting reference, see page 6 line 144.

We agree that the sentence in the original manuscript may imply implementation of the full MLCC model in these maternity care networks. Therefore, we clarified that these networks implemented an MLCC-enhancing innovation, see page 6 line 148. In addition, we explained the aim of those MLCC-enhancing innovations in relation to the continuity of care context; providing an opportunity for women to continue receiving care from their own known midwifery team instead of a complete transfer to secondary care, see page 6 lines 149-151.

We have removed the comma.

We agree that this addition is essential to describe in the data collection process. Therefore, we have added how participants were invited (by email, telephone or in person). The interviews were conducted and recorded via Microsoft Teams and all conducted by the first two authors, see page 8 line 181-182.

We have added a translated interview guide as a supplemental material to the manuscript. We refer to this supplement on page 8 line 183.

Results

The sequence of the Results has frequently been revised and discussed among the authors. We believe that starting with the NPT constructs is in line with the aim of the study and builds towards the overarching themes. However, we agree that it can be helpful to address these overarching themes shortly in the beginning of the Results to enhance the logic structure of the manuscript and to provide the reader with some narrative flow. Therefore, we relocated lines 334-335 from the original manuscript to before the NPT constructs, including the financial aspect, together with Figure 1, see page 11 lines 222-223.

We have added the NPT constructs before each theme, see page 11 line 227, page 12 line 248, page 13 line 277, and page 15 line 306.

Discussion

In our manuscript, the “Implications for practice” is intentionally presented as a sub-section within the broader Discussion, rather than as a separate section. We have added the subheadings solely for readability. For that reason, we have kept these parts distinct in structure but connected in content.

Strengths and limitations

We have added the limitation of not being able to examine the implementation of a full MLCC model because this appears to be difficult in practice, see page 23 lines 509-510.

We have added “of” before trust, see page 22 line 483.

Conclusion

As stated in our previous response, we have now clarified throughout the manuscript that our study focusses on MLCC-enhancing initiatives, rather than full implementation of MLCC. This has also been clarified in the conclusions, see page 25 line 568.

Reviewer 2

Thank you for seeing the value of our work and helping us to improve our manuscript with your valuable comments and suggestions.

We have used the Framework Method as an overarching analytic structure, with the four core constructs of the NPT functioning as our analytical framework. Within this structure, we remained open to identifying additional themes that emerged from the data. In our study, NPT served as the organizing framework rather than a matrix in the traditional sense of the Framework Method, so we did not construct a full matrix as described in Gale et al. (2013). Our coding process did involve systematically indexing data to these constructs, as well as to inductively derived themes. We believe this is now more clear with the additions we have made on page 8 lines 189-190.

Methods

We agree that the original version of the manuscript could give the impression that full MLCC models had been implemented in the included networks. Based on your comment, we have clarified throughout the manuscript that the study concerns innovations aimed at increasing continuity of care, rather than full implementation of MLCC. In addition, we changed the title and have added a sentence to the introduction to state that an immediate implementation of a complex intervention like MLCC is not feasible. The present description clarifies that the included networks did not achieve full MLCC, but rather took steps toward improving continuity.

Please see page 2 lines 31 and 49, page 3 lines 72-74, page 4 line 92, page 6 line 148, page 7 line 157, page 22 lines 488 and 497, and page 25 line 568.

Moreover, we have reformulated the sentence about “networks that appeared to come close to achieving full continuity of care”, see page 7 lines 160-162. Those networks made broader organizational changes to achieve a higher degree of continuity.

Results

We agree that the term ‘potential’ is misplaced in this context. It has been removed, see page 18 line 393.

We agree that the sequence of the Results is challenging. It has frequently been revised and discussed among the authors. To enhance the logic structure of the manuscript we have added a short description of the Results before the NPT constructs are described, see page 11 lines 222-223. Furthermore, we relocated Figure 1 to the beginning of the Results section.

In order to conceptualize the relationships of the core constructs and the central overarching themes trust and financial feasibility we have moved Figure 1 to the beginning of the Results section.

Discussion

We have added that ‘trust’ refers to as ‘trust between stakeholders’ in this context, see page 20 line 453, and page 25 line 570.

---

## [Decision Letter · Decision Letter 1]

12 Jan 2026

Understanding the implementation of continuity-enhancing innovations as steps towards midwife-led continuity of care: a qualitative study using Normalization Process Theory

PLOS One

Dear Dr. Simmelink,

Thank you for submitting your manuscript to PLOS ONE. After careful consideration, we feel that it has merit but does not fully meet PLOS ONE’s publication criteria as it currently stands. Therefore, we invite you to submit a revised version of the manuscript that addresses the points raised during the review process.

We look forward to receiving your revised manuscript.

Kind regards,

Arne Johannssen

Academic Editor

PLOS One

Journal Requirements:

Additional Editor Comments:

Please carefully address the comments raised by both reviewers.

Reviewers' comments:

Reviewer's Responses to Questions

**Comments to the Author**

Reviewer #2: (No Response)

Reviewer #3: (No Response)

2. Is the manuscript technically sound, and do the data support the conclusions?

Reviewer #2: Yes

Reviewer #3: No

3. Has the statistical analysis been performed appropriately and rigorously?

Reviewer #2: N/A

Reviewer #3: N/A

4. Have the authors made all data underlying the findings in their manuscript fully available?

Reviewer #2: No

Reviewer #3: No

5. Is the manuscript presented in an intelligible fashion and written in standard English?

Reviewer #2: Yes

Reviewer #3: Yes

Reviewer #2: Thank you for the opportunity to read the revised version of your manuscript. It is evident that you have invested considerable effort in addressing the previous comments, and several sections have been strengthened as a result. I do, however, have one remaining concern that I do not feel has been fully resolved: the positioning of your findings in relation to the existing body of research.

While the discussion section now includes reflections on clinical implications, as well as strengths and limitations, the subsection that situates your results within the context of prior studies remains rather limited. If this brevity reflects a genuine lack of previous research in the area, I encourage you to state this explicitly. Otherwise, expanding this part of the discussion would help readers better understand how your findings align with, diverge from, or extend what is already known.

Regarding line 465, I would like to draw your attention to what appears to be a typographical error. Did you intend to write “country” rather than “county”? Clarifying this would help avoid potential misunderstanding.

In addition, I am still unable to locate the completed SRQR checklist. If it has been uploaded, it may be helpful to indicate its exact location in the submission system. If it has not yet been provided, please ensure that the filled checklist is included with the revised materials.

Reviewer #3: 11.01.2026

Reviewer’s Comments on the Manuscript: “Understanding the implementation of midwife-led continuity of care using the Normalization Process Theory”

Thank you for the opportunity to review this interesting and relevant paper. The manuscript addresses an important gap in the literature. Below are my detailed comments.

Abstract

The abstract should be revised after incorporating changes to the main manuscript based on reviewer recommendations.

Background

The background section would benefit from clearer definitions and explanations. Currently, the understanding of MLCC, its strengths, and barriers to implementation is left largely to the reader. Please clarify:

• What exactly are Maternity Care Networks?

• What is the MLCC model within these networks?

• How is MLCC embedded in the referral system?

• Who is involved at the point of referral?

• What are the expected outcomes of MLCC?

Methods

Setting, Participants, and Data Collection

Information about Maternity Care Networks is difficult for readers unfamiliar with the context. Consider moving this information to the background section.

Additionally:

• Explain who was recruited and how recruitment occurred within these networks.

• Provide a sampling matrix to illustrate the purposive sampling strategy. This will help readers understand purposive sampling strategy and will be useful when presenting results.

Data Analysis

The description of the Thematic Framework approach does not align with corresponding methods. Please include the following steps if this approach was used:

(1) Transcription; (2) Familiarisation; (3) Coding; (4) Developing an analytic framework; (5) Indexing; (6) Charting; and (7) Interpreting the data.

Results

Participant Characteristics

Clarify participant characteristics within the context of Maternity Care Networks. This links back to the need for a sampling matrix and details on theoretical sampling.

Overview of Results

The emerging framework within NPT is not presented. An illustration at the beginning of the results section showing the themes and their interpretation in the MLCC context would greatly improve clarity. Currently, the relationship between themes and the NPT framework is difficult to follow.

Discussion

The discussion lacks structure and does not align with SRQR guidelines for integrating results with existing evidence. This section requires substantial revision to ensure clarity and coherence.

.

Reviewer #2: No

Reviewer #3: No

---

## [Author Response · Author response to Decision Letter 2]

22 Feb 2026

We understand the potential confusion in the initial version of the discussion, where several statements summarizing our results were presented in a general manner without indicating that they stemmed from our own findings. We have now revised this section to explicitly attribute these points to our study results, see lines 463-490. Furthermore, we have expanded the ‘Implications for practice’ section by integrating additional relevant literature, and situating our findings more clearly in relation to prior research and highlighting how our study extends existing knowledge, see lines 529-598.

Thank you for your attentiveness, it should be indeed “country”.

The SRQR checklist had indeed not been included in the original submission, but has now been added.

Reviewer 3

After the revisions, we checked and updated the abstract to ensure that it still accurately reflects the content of the revised manuscript.

We acknowledge the challenge of explaining the Dutch system clearly in an international context. We have added a textbox within the introduction for further clarification. In doing so, we outline a more comprehensive picture of the organization of maternity care in the Netherlands. To give a better understanding of the scope and structure of a maternity care network, we have included additional explanations, see lines 121-126:

“Maternity care networks are a formal regional partnership between a hospital and the surrounding community-based midwifery practices. Within a maternity care network, hospital-based care providers and community midwives work closely together to coordinate maternity care, ensuring clear referral pathways, and shared responsibility throughout pregnancy, childbirth, and the postnatal period. Community-based midwifery practices may be affiliated with more than one maternity care network.”

Women who develop an intermediate or high risk for complications are referred to obstetrician-led hospital care. This referral is initiated by community midwives and has implications for the continuity of care (see lines 101-105):

“Women who develop an intermediate or high risk for complications are referred to obstetrician-led hospital care, often resulting in a change of care professionals. Such referral systems can disrupt continuity of care, as women often need to transition between different care settings and professionals throughout their pregnancy, during childbirth, or in the postpartum period.”

We elaborate on the expected outcomes of MLCC in lines 64-69.

Methods

To clarify the structure of maternity care networks we added additional background information within the textbox “organization of maternity care in the Netherlands” (see lines 121-126).

In response to your first comment, we have expanded the Methods section to more clearly explain who was recruited and when we stopped sampling. Recruitment is explained in lines 178-188:

“Participants were recruited through an initial purposive sampling strategy. The first participants invited were members of regional MLCC implementation project groups, as they were directly involved in the design and implementation of the innovation. As data collection progressed, theoretical sampling was applied, meaning that additional participants were selectively invited based on emerging findings from earlier interviews. By iteratively refining participant selection, key perspectives and experiences relevant to the implementation of MLCC were deeper explored. Sampling continued until sufficient information had been obtained within each region to capture experiences relevant to the constructs of NPT, and no substantively new insights emerged in subsequent interviews. Participants were invited by email, telephone or in person. Semi-structured interviews were conducted and recorded via Microsoft Teams by RS and AP, using a topic list based on sensitizing concepts considered relevant in literature (supplement material 2).”

We have revised the tables in the Results section to provide a clear overview of all participants per region. These tables are intended to illustrate the sampling strategy and to support transparency regarding the composition of the sample across regions. We believe this provides readers with a clear overview of the sampling approach and facilitates interpretation of the findings. See table 1 and table 2 on page 11.

We agree that greater clarity regarding the analytical steps of the Thematic framework is warranted. All stages of the Framework method were undertaken. However, in our study, we applied the Framework method in a theory-informed manner, using the four core constructs of NPT as an a priori analytical framework. This reflects a predominantly deductive approach, while remaining open to inductively generated themes. We have now revised this part of the Method section to clearly describe the analytical steps and to further clarify how the NPT framework guided analysis, see lines 194-210.

Results

As stated above, we have revised the tables in the Results section to provide a clear overview of participant characteristics per region. Moreover, we expanded the section describing the sampling method for greater clarity.

The themes presented in the results correspond to the core constructs of NPT. During the analysis, it became apparent that two themes (financial feasibility, and trust) emerged within each construct. For this reason, we elected to describe them separately. To clarify more that the themes correspond to the core construct of NPT we added figure 1 at the beginning of the Results section and named the NPT construct at the beginning of each paragraph (coherence, cognitive participation, collective action, reflexive monitoring). We inserted a clarifying remark to further elucidate this, see line 238-239:

“Trust and financial feasibility emerged as central and cross-cutting themes overarching the four core constructs of NPT, influencing the implementation and normalization of MLCC (see Figure 1).”

Discussion

We have separated our Discussion section into a summary of our main findings, strengths and limitations, and implications for practice. We understand the potential confusion in the initial version of the discussion, where several statements summarizing our results were presented in a general manner without indicating that they stemmed from our own findings. We have now revised this section to explicitly attribute these points to our study results, see lines 463-490. In the implications for practice section, we explicitly integrated our results with relevant prior literature, highlighting where our findings align with, extend, or differ from existing evidence, see lines 529-598. We believe that the discussion section now aligns with the SRQR guidelines.

---

## [Decision Letter · Decision Letter 2]

6 Mar 2026

Dear Dr. Simmelink,

Thank you for submitting your manuscript to PLOS ONE. After careful consideration, we feel that it has merit but does not fully meet PLOS ONE’s publication criteria as it currently stands. Therefore, we invite you to submit a revised version of the manuscript that addresses the points raised during the review process.

Please carefully consider the reviewer's suggestions.

We look forward to receiving your revised manuscript.

Kind regards,

Arne Johannssen

Academic Editor

PLOS One

Journal Requirements:

Reviewer's Responses to Questions

**Comments to the Author**

Reviewer #1: (No Response)

Reviewer #2: All comments have been addressed

2. Is the manuscript technically sound, and do the data support the conclusions?

Reviewer #1: Yes

Reviewer #2: Yes

3. Has the statistical analysis been performed appropriately and rigorously?

Reviewer #1: N/A

Reviewer #2: N/A

4. Have the authors made all data underlying the findings in their manuscript fully available?

Reviewer #1: No

Reviewer #2: No

5. Is the manuscript presented in an intelligible fashion and written in standard English?

Reviewer #1: Yes

Reviewer #2: Yes

Reviewer #1: Thanks for giving me the chance to review this paper again. You have done a great job improving it; however, I have some minor concerns about the Discussion section. Please see my specific comments in the attachment.

Reviewer #2: Thank you for your careful consideration of my previous comments and for providing suggested revisions.

I still cannot locate the filled SRQR checklist. Although the authors state that it has been uploaded, it does not appear in the PDF provided for review. It is possible that this may be related to the submission system.

I also believe that the manuscript would benefit from being shorter and more concise; however, if there are no word limits, this may ultimately be a matter of preference. After reading the full Results section, I find it difficult to identify the main findings of the study. While the conclusion and abstract are clear on this point, I am uncertain whether all the information presented across nine pages of results is necessary for the readers.

In the Discussion section, lines 456–481 appear to continue the presentation of results rather than offering interpretation.

To strengthen the manuscript, you may wish to revise this section to connect it with the contextualisation of your findings in relation to existing literature.

Moreover, I am unfamiliar with a strengths‑and‑limitations section in a qualitative study that does not address aspects of trustworthiness. Consider elaborating on credibility, dependability, confirmability, or transferability to enhance the rigor of the manuscript.

.

Reviewer #1: No

Reviewer #2: No

---

## [Author Response · Author response to Decision Letter 3]

25 Mar 2026

Reviewer 1

We have revised the sentence accordingly and removed the comma and ‘were’ to improve the language, see page 2 line 37.

We have added the word ‘elements’ to make clear that not a full MLCC model was implemented, see page 20 line 455.

We have revised the structure of the Discussion section. The subsection ‘Strengths and Limitations’ has been moved to the end of the Discussion, following the ‘Implications for Practice’ subsection, so that the summary of results and integration of our findings with existing literature remain grouped together.

We have retained a brief paragraph summarizing the main findings before integrating them with the existing literature. This structure is commonly used in qualitative research articles to orient the reader to the key findings before engaging in interpretation and comparison with prior work. Similar Discussion structures can be found in other articles published in this journal, where the Discussion likewise begins with a short summary of the findings before engaging with the literature. See for example: The use of international comparison as interactive teaching method in pharmacy education | PLOS One. However, the opening paragraph has been shortened by removing several sentences, in order to improve readability, see pages 20&21 lines 455-480.

Reviewer 2

The completed SRQR checklist was uploaded during submission. It is therefore possible that this issue is related to the submission system. To ensure accessibility for the reviewers, we have uploaded the checklist again with the revised submission.

In response to this suggestion, we have shortened the Results section by removing several explanatory sentences while preserving the core findings.

In line with this comment, we have shortened the opening paragraph of the Discussion section by removing several sentences. We choose to retain a brief paragraph summarizing the main findings, which is more commonly used in qualitative research. Similar Discussion structures can be found in other articles published in this journal, where the Discussion likewise begins with a short summary of the findings before engaging with the literature. We have, however, strengthened the connection with existing literature in the subsequent paragraphs, by relocating the ‘Strengths and Limitations’ section to the end of the Discussion.

We have expanded the Strengths and Limitations section to more explicitly address aspects of trustworthiness in qualitative research, including credibility, confirmability, and transferability. On page 26 lines 604-607 we added:

‘Credibility was further strengthened through the inclusion of a wide range of stakeholders across different professional groups and maternity care networks, enabling triangulation of perspectives. In addition, the multidisciplinary composition of the research team facilitated critical reflection, thereby enhancing confirmability of the findings.’

Furthermore, we made additions in lines 610 and 621.

---

## [Decision Letter · Decision Letter 3]

8 Apr 2026

Understanding the implementation of continuity-enhancing innovations as steps towards midwife-led continuity of care: a qualitative study using Normalization Process Theory

PONE-D-25-40430R3

Dear Dr. Simmelink,

We’re pleased to inform you that your manuscript has been judged scientifically suitable for publication and will be formally accepted for publication once it meets all outstanding technical requirements.

Kind regards,

Arne Johannssen

Academic Editor

PLOS One

Additional Editor Comments (optional):

Reviewers' comments:

Reviewer's Responses to Questions

**Comments to the Author**

Reviewer #1: All comments have been addressed

2. Is the manuscript technically sound, and do the data support the conclusions?

Reviewer #1: Yes

3. Has the statistical analysis been performed appropriately and rigorously?

Reviewer #1: N/A

4. Have the authors made all data underlying the findings in their manuscript fully available?

Reviewer #1: No

5. Is the manuscript presented in an intelligible fashion and written in standard English?

Reviewer #1: Yes

Reviewer #1: Thank you for allowing me to review this paper again. I am satisfied with the revisions made in response to the reviewers’ comments and believe the paper is ready for acceptance.

.

Reviewer #1: No

---

## [Editor Report · Acceptance letter]

PONE-D-25-40430R3

PLOS One

Dear Dr. Simmelink,

I'm pleased to inform you that your manuscript has been deemed suitable for publication in PLOS One. Congratulations! Your manuscript is now being handed over to our production team.

Kind regards,

on behalf of

Profesor Arne Johannssen

Academic Editor

PLOS One